# Indium and Antimony Distribution in a Sphalerite from the "Burgstaetter Gangzug" of the Upper Harz Mountains Pb-Zn Mineralization

**Thomas Schirmer [1],\*, Wilfried Ließmann [1], Chandra Macauley [2,3] and Peter Felfer [2]**

1    Institute of Disposal Research, Clausthal University of Technology, Adolph-Roemer-Str. 2A,
D-38678 Clausthal-Zellerfeld, Germany; wilfried.liessmann@tu-clausthal.de
2    Materials Science & Engineering, Institute I, Friedrich-Alexander-Universität Erlangen-Nürnberg (FAU),
Martensstr. 5, 91058 Erlangen, Germany; chandra.macauley@fau.de (C.M.); peter.felfer@fau.de (P.F.)
3    Interdisciplinary Center for Nanostructured Films (IZNF), Cauerstraße 3, 91058 Erlangen, Germany
\*    Correspondence: thomas.schirmer@tu-clausthal.de; Tel.: +49-5323-72-2917

**Abstract:** The sphalerite from the Burgstaetter Gangzug, a vein system of the Upper Harz Mountain nearby the town of Clausthal-Zellerfeld, exhibits a very interesting and partly complementary incorporation pattern of Cu, In and Sb, which has not yet been reported for natural sphalerite. A sphalerite specimen was characterized with electron probe micro-analysis (EPMA) and atom probe tomography (APT). Based on the EPMA results and a multilinear regression, a relation expressed as Cu = 0.98In + 1.81Sb + 0.03 can be calculated to describe the correlation between the elements. This indicates, that the incorporation mechanisms of In and Sb in the structure differ substantially. Indium is incorporated with the ratio Cu:In = 1:1 like in roquesite ($CuInS_2$), supporting the coupled substitution mechanism $2Zn^{2+} \rightarrow Cu^+ + In^{3+}$. In contrast, Sb is incorporated with a ratio of Cu:Sb = 1.81:1. APT, which has a much higher spatial resolution indicates a ratio of Cu:Sb = 2.28:1 in the entire captured volume, which is similar to the ratio calculated by EPMA, yet with inhomogeneities at the nanometer-scale. Analysis of the solute distribution shows two distinct sizes of clusters that are rich in Cu, Sb and Ag.

**Keywords:** sphalerite; antimony; indium; element distribution; EPMA; APT; Upper Harz Mountains; Pb-Zn-mineralization

## 1. Introduction

### 1.1. Overview and Motivation

As elements like indium (In) and antimony (Sb) are essential for production of light-emitting diodes (LEDs) and laser diodes (in InGaN or InGaP), thin films for liquid-crystal displays (LCD, in indium tin oxide (ITO)) or for semiconductor applications (In, Sb), the sources of these elements are becoming more and more important. Therefore, an understanding of the geochemistry of these elements is increasingly of interest. During the last years in the Harz Mountain region, efforts have been made to assess the availability of In in tailings of the Rammelsberg mine site. One topic of the investigations was the incorporation, concentration and distribution of In within the different mineral compounds of these tailings (pyrite, chalcopyrite, sphalerite, galena, barite). Sphalerite turned out to be the most promising candidate for In-enrichment (see Section 1.2).

Due to the limited concentration range of In within the tailing of the Rammelsberg mine site, the investigations were extended to indium-rich sphalerites of upper Harz Mountain ore formations, among others also the "Burgstaetter Gangzug" (see Section 1.2). The element distribution patterns of Cu, In and Sb within these sphalerites are presented in this work based on electron microprobe microanalysis (EPMA) and atom probe tomography (APT).

## 1.2. Overview of the Geological Setting of the Ore Body

In the northwestern part of the Harz mountains there are WNW-ESE ("hercynic") striking fault zones, normally steeply inclined to south, that are 19 to 20 km long. Eight of them, in the central and the western part of the district, were of economic value. Hydrothermal mineralization occurs as separated oreshoots with high metal contents and complex mineralogical composition and structures. Most of the big and high-grade ore bodies are bound to horsetail-structures, where a sterile fault is divided into several different ore veins and veinlets, forming a network-like system.

The width of the fault zones varies from a few meters up to 100 m. The thickness of the mineralized veins, filled with fragmented country rock, often hydrothermal altered to loam, gangue minerals (quartz, calcite, locally siderite and barite) and more or less sulphides, reach up to 10 m. Country rocks are mainly graywackes and slates of lower carboniferous age.

Main sulphides are galena, sphalerite, chalcopyrite, pyrite and less tetrahedrite, in varying proportion, coarse grained intergrown with the gangue minerals. Typical are brecciated, vertical banded, ring- and cockade shaped textures [1,2]. Analyses of galena from different places of the Wilhelmer orebody reveal concentrations of 0.004%–0.17% Ag and 0.17%–0.97% Sb [2]. Main Sb-phases are bournonite and tetrahedrite, indicating the Sb-containing environment of the sampled ore.

The "Burgstaetter Gangzug" vein system of the Upper Harz Mountain nearby the town of Clausthal-Zellerfeld is well examined and represented in a paragenetic scheme [3]. According to this scheme, the sample belongs to the first main stage of the mineral emplacement (substage IIb, "derbe Zinkblendetrümer"), which is, in this case, very rich in quartz. This is due to a Mesozoic event, dated in the interval Upper Triassic to the border of the Triassic Jurassic (226–200 my) period [1]. Of particular interest for samples taken from this site, is the availability of Sb, which enters the crystal lattice of sphalerite. Unfortunately, this portion of the mine is now flooded and therefore no longer accessible, making the samples investigated in this study rather unique.

## 1.3. Short Review of In and Sb Incorporation in Sphalerite

Since sphalerite is an important host of In [3,4], the incorporation of In into sphalerite as well as the responsible substitution process is the subject of many studies.

The In concentration of most natural sphalerite is below 600 μg/g, [5–13]. However, if a monovalent element such as Cu ($Cu^+$) is present during crystallization of sphalerite, the enrichment of In can be distinctively higher. Additionally, Cu and In concentrations in sphalerite mostly show a positive correlation [14]. Therefore, besides dedicated In minerals like roquesite, high In enrichment in sphalerite occurs mostly together with incorporation of copper (Cu). Because the solubility of Cu in sphalerite is not very high (<5000 μg/g, [15]) and a substitution of $Zn^{2+}$ with $Cu^{2+}$ is unlikely [16], the relation of the Cu and In concentration in sphalerite leads to the theory of a coupled substitution mechanism. Due to the described coupled substitution mechanism (Equation (1)) the concentrations of the monovalent (Cu, Ag) and trivalent cations (Ga, In, Sb) and even tetravalent cations (Ga, Sn) can be correlated.

$$4Zn^{2+} \leftrightarrow X^+ + Y^{3+} + Z^{4+} \tag{1}$$

In this general equation, X represents monovalent elements like Cu or Ag, Y represents In, Sb and Z represents Ga or Sn. Data with correlations of this type is published by various authors (e.g., [7,10,13,14,17–20]).

The ratios of substituted Cu and In are similar to those of roquesite ($CuInS_2$) or a sakuraiite-like mineral with the formula $CuZn_2InS_4$ (e.g., [21,22]) with a ratio of Cu:In = 1:1. A sphalerite deposit forming large discrete grains with distinct Fe/Zn zoning that exhibits two correlations of In and Cu with 1Cu: 1In (as in roquesite) and 3Cu: 1In is reported by [18]. The latter ratio occurs together with traces of Sn (as in sakuraiite, $(Cu,Zn,Fe)_3(In,Sn)S_4$; e.g., [3]).

The incorporation of In and Cu on the basis of roquesite is plausible, because, to a certain extent, solid solutions can form along the line roquesite and sphalerite. This line is not continuous. There exists complete miscibility between $CuZn_2InS_4$ and sphalerite (e.g., [23,24]). In contrast, the miscibility of $CuZn_2InS_4$ and roquesite is limited [24]. A similar miscibility gap in the roquesite-rich area of that line was determined by [25]. In seems to be incorporated directly into the sphalerite lattice and does not form nano-crystallites (e.g., [20]) and references therein), although a submicroscopic phase with Ge in chalcopyrite is proposed by [26].

Indium concentrations in copper-containing sphalerite can vary in the range of 1500 to 10,000 µg/g (e.g., [14,27–30]), up to 6–7 wt.% (e.g., [22,28]) or even 15–20 wt.% (e.g., [12,20]).

In contrast to In, Sb enrichment is not often reported in sphalerite. Sb incorporation into ZnS is investigated by [31] in acidular sphalerite from the Kokanee range, British Columbia, Canada. In these samples, Sb is incorporated along crystallographic planes with a maximum concentration of 0.55 wt.%. It is assumed, that the ZnS (as wurtzite) was precipitating from hydrothermal fluids with elevated Cu, Fe, Ag and Sb concentration, followed by precipitation of sphalerite from depleted solutions. Later the wurtzite was inverted to sphalerite. Antimony concentrations of 0.4–2400 µg/g were found in sphalerite from Red Dog, Alaska by [32]. Correlation between Sb and Cu is published by [33] in North Pennine sphalerite formed from a saline fluid at maximum temperature of 140 °C. The Sb concentrations range from 0.05–0.15 wt.%. A very weak positive correlation of In and Sb is reported by [8] with a concentration range of about 1–1000 µg/g in ores from the Dulong and Dachang mines, South China.

## 2. Material and Methods

### 2.1. Material

The specimen, taken from the minerals collection of the TU Clausthal was sampled approximately in 1920, ca. 800 m below the surface in the center of the main orebody. The modal composition of the sample is estimated to consist of 85 wt.% coarse grained, partly idiomorphic quartz, 5 wt.% calcite, 3 wt.% sphalerite, 3 wt.% galena and 3 wt.% chalcopyrite. The cockade-shaped sulfides are concentrated near the sidewall of the vein, which is slickensided black slate.

The investigated coarse-grained sphalerite (Figure 1) is dark brownish and intergrown with quartz, galena and chalcopyrite. This ore is characterized by complex twinning and zoning of the crystals. The sphalerite grains are bulky and contain no exsolution of chalcopyrite. The size of the crystals is up to ~4.8 mm (Figure 1). For sphalerite from the Burgstaetter vein system, [2] published average contents of 2.85–4.35 wt% Fe and wt% 0.4–0.7 % Cd. Reliable data on other trace elements are not available.

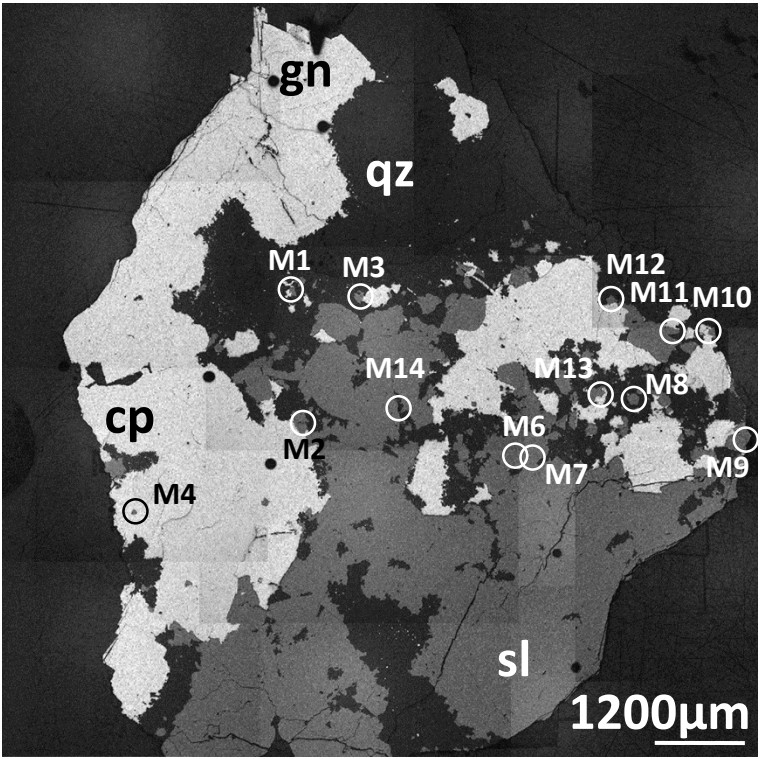

**Figure 1.** Reflected light micrograph of the complete sample. The positions of all element distribution mappings marked with a white circle, cp—chalcopyrite, gn—galena, sl—sphalerite, qz—quartz.

*2.2. Methods*

EPMA is a standard method to characterize the chemical composition of mineral crystals in terms of single spot analysis or element distribution patterns, accompanied by electron backscattered (BSE) or secondary electron (SE) micrographs. To carry out EPMA measurements, the sample was prepared as polished block in epoxy resin, coated with carbon and characterized using a Cameca SXFIVE FE (Field Emission) electron probe (Cameca, Gennevillier, France), equipped with five wavelength dispersive (WDX) spectrometers. For optimal detection sensitivity, the acceleration voltage was set to 25 kV at 100 nA, delivering a substantially better peak/background ratio and lower heating of the sample surface due to lower electron surface density at the same current (nA) setting compared to low voltage (kV) settings.

For better comparison, L-lines were used for all elements (except for S). The beam size was set to 0, leading to a beam diameter of substantially below 1 μm (100–600 nm with field emitters of Schottky-type, e.g., [34]). To evaluate the measured intensities the X-PHI-Model was applied [35].

To collect the element distribution profiles, In and Sb (both Lα) were measured on three wavelength dispersive spectrometers (WDX) equipped with two large (L)PET (pentaerithritol) and one standard PET crystal (Table 1). The count rates (peak and background) for In and Sb were averaged over all spectrometers. Cu was determined with Lα, using a standard TAP (thallium hydrogen phthalate) crystal. The minor elements Fe (using the Ll instead of the Lα because of better reproducibility) and Cd (Lα) were determined using a large (L)TAP (thalliumphtalat) crystal. To correct for the matrix, the matrix elements Zn (Lα) and S (Kα) were screened with rather short measurement time using a standard TAP and a large (L)TAP, respectively. The measurement times were selected such that the highest accuracy was achieved for In and Sb, followed by Cu. For comparable analysis depth, all elements, except for S, were analyzed using L-Lines. The element distribution mappings for Cu, Cd, In and Sb were collected on the WDX spectrometers using the same X-Ray lines and analytical settings (kV, nA, crystal). The images were recorded with 10 ms dwell time and a step width of

100 nm. The (semi)quantification of the element distribution mappings using the same X-Ray lines was carried out with the reference measurements, used for calibrating the element distribution profiles. The reference materials were provided by P&H Developments, The Shire 85A Simmondley village, Glossop, Derbyshire, UK.

**Table 1.** Measurement parameters for setting up the electron probe micro analyzer. Bkg = background measurement, DL—detection limit (calculated by instrument software), OVL—critical line overlap, Ref—international reference, used for calibration.

| | Crystal | Peak (s) | $Bkg_1$ (s) | $Bkg_2$ (s) | DL 25kV, wt.% | Ref |
|---|---|---|---|---|---|---|
| $SK\alpha$ | LPET | 20 | 10 | 10 | 0.02 | ZnS |
| FeLl * | LTAP | 60 | 10 | 10 | 0.49 | $FeS_2$ |
| $CuL\alpha$ | TAP | 180 | 90 | 90 | 0.01 | $CuFeS_2$ |
| $ZnL\alpha$ | TAP | 60 | 10 | 10 | 0.03 | ZnS |
| $CdL\alpha$ | LPET | 30 | 15 | 15 | 0.01 | CdS |
| $InL\alpha$ | PET/LPET/LPET | 240/204/204 | OVL $CdL\beta$ | 120/102/102 | 0.004 | InSb |
| $SbL\alpha$ | PET/LPET/LPET | 240/204/204 | 120/102/102 | 120/102/102 | 0.004 | InSb |

\* For Fe the Ll-line instead of the $L\alpha$ was used.

APT is a technique that can be used to produce single atom, 3D maps of solids materials, in a small volume of several 10 s to several 100 s of nanometers in size. This is achieved by field ionizing and field evaporating single ions off of a sharp, needle-shaped sample. The ions are then captured on a 2D, time resolved detector that allows for the identification of the ion via its time-of-flight [36]. The use of a laser to trigger the departure of the ions is required to provide a reference time for the flight time measurement. In metallurgy, often high voltage pulses are used to trigger field evaporation but this requires conductive samples and is thus not applicable to minerals.

APT has already been successfully applied in geology and planetary sciences for example, to confirm or enhance dating [37], or assess trace element redistribution in minerals [38]. The needle-shaped sample for atom probe analysis was prepared on a ZEISS Crossbeam 540 focused ion beam/secondary electron microscope (FIB/SEM, Oberkochen, Germany), using the standard site-specific lift out procedure. A suitable location (M3 in Figure 1) was first identified with EPMA to ensure the atom probe sample was taken from a Sb-rich region. Using a high electron current (20 nA) and voltage (25 kV), the desired location was found using backscatter electron imaging, Figure 2a) in the FIB/SEM. A pillar, ~1 μm in diameter, was milled using a 3 nA, 30 kV Ga-ion FIB, Figure 2b), removed with a micromanipulator and attached to a tungsten substrate using the platinum gas injection system (GIS) in the FIB/SEM, Figure 2c). The pillar was then sharpened with 100 pA, 10kV to the required tip-shape for atom probe. The APT experiments were conducted on a CAMECA LEAP 4000X HR instrument equipped with an ultraviolet laser.

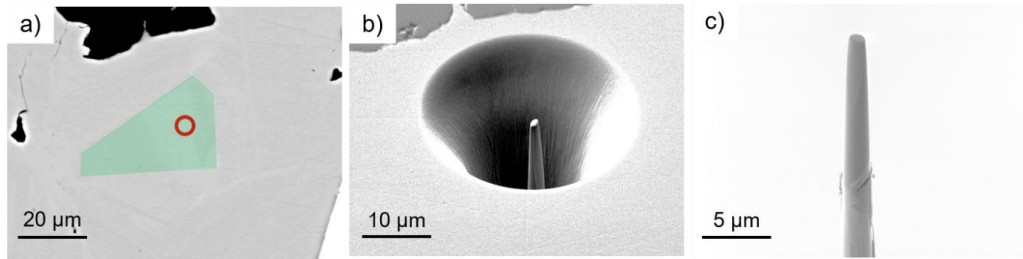

**Figure 2.** (**a**) A backscatter electron image (25 kV, 20 nA) shows the Sb-rich region (highlighted in green) from which a pillar was milled using the Ga-focused ion beam (FIB) from at the position marked with a red-circle. The pillar, (**b**) was removed from the bulk material using a micromanipulator and (**c**) attached to a tungsten substrate using the platinum gas injection system in the focused ion beam/secondary electron microscope (FIB/SEM).

Data was acquired in laser pulsing mode at a specimen temperature of 60 K, a laser pulse energy of 90 pJ, a pulsing rate of 200 kHz and a target evaporation rate of five ions per 1000 laser pulses. The detection system of the used instrument consists of a mass-resolution enhancing electrostatic mirror reflectron [36] and a crossed delayline detector with standard microchannel plates as the pre-amplification. This yields a detection efficiency of around 37% of the emitted ions that are projected onto the detector. As a result, formations down to about 30 atoms can be effectively analyzed [39].

The APT dataset was reconstructed using a backprojection algorithm published in [40]. The precipitation of small clusters was quantified using a method based on Voronoi cell volumes, published in [41].

## 3. Results

Compositional results determined by EPMA are presented first, followed by the APT characterization. The element composition was assessed via EPMA through a qualitative peak search throughout the sample and the following elements were found in significant concentrations (within the limits of the analytical method, Table 1): S, Fe, Cu, Zn, Cd, In and Sb. Other trace elements like Ga, Ge or Sn could not be detected.

Significant amounts of Ag were only detected in tetrahedrite inclusions in galena, which are beyond the scope of this study. On assessment of the element distribution, the Cu, Cd, In and Sb concentrations within the sphalerite showed significant correlations.

Because the data were generated using an automated line scan, exceptionally low totals could result from measurements on fissures or cracks in the sphalerite crystal. The overall lower totals may result from the fact that Zn and S were only screened with short measurement times to carry out matrix correction. Additionally, Fe and Zn were analyzed L$\alpha$-Lines with inferior analytical precision (compared with the K$\alpha$-Lines).

The enrichments of Cu, Cd, In and Sb can be found in all parts of the sphalerite crystals. There is no increase of the Cu or Ag concentration directly at the contact to the chalcopyrite or galena grains. This renders direct contact of the crystals as well as element diffusion into the sphalerite during formation improbable. Chalcopyrite and galena itself are virtually free of Cu, Cd, In and Sb. The concentration ranges in sphalerite are 0.46–2.14 at.% Fe, 0–0.7 at.% Cu, 0.1–0.48 at.% Cd, 0.01–0.52 at.% In and 0–0.39 at.% Sb.

This section may be divided by subheadings. It should provide a concise and precise description of the experimental results, their interpretation as well as the experimental conclusions that can be drawn.

### 3.1. Compilation of All Element Distribution Profiles

To investigate the overall correlations for all point analyses of the measured elements, the atomic percent (at.%) values for a number of element distribution profiles (n = 280) in the sphalerite crystals were plotted in one xyz diagram (Figure 3a). Whilst the correlations Cu/In, Cu/Sb and In/Sb are not very pronounced, a plane can be identified resembling a more complex multi linear correlation in the xyz diagram (Figure 3a,b).

With multilinear regression, the following formula can be calculated:

$$Cu = 0.988In + 1.81Sb + 0.03 \tag{2}$$

This leads to a ratio of Cu:In = 1:1 and Cu:Sb = 1.81:1 (Equation (2)). A clear positive correlation of element pairs is only apparent for Cu/Sb (Figure 3c). With Cu/In there is no clear positive correlation but merely a "forbidden zone" in the lower right corner of the x/y diagram (Figure 3d). In and Sb do not have a clearly defined correlation (Figure 3e). Additionally, a ternary plot reveals a general negative trend of Cd and Cu/(In + Sb) (Figure 4).

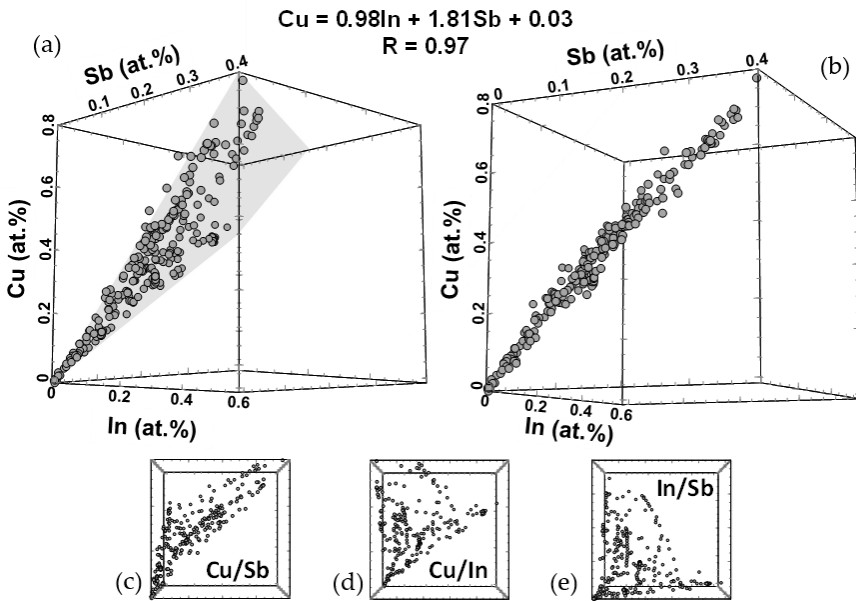

**Figure 3.** 3D-plot of the whole Cu, In and Sb element distribution profiles dataset (n = 280). The three x/y diagram components are displayed below. For explanations, see text. (**a**) Overall 3D plot with plane; (**b**) Cross section of overall 3D plot;(**c**) Cube side Cu(y)/Sb(x); (**d**) Cube side Cu(y)/In(x); (**e**) Cube side Sb(y)/In(x).

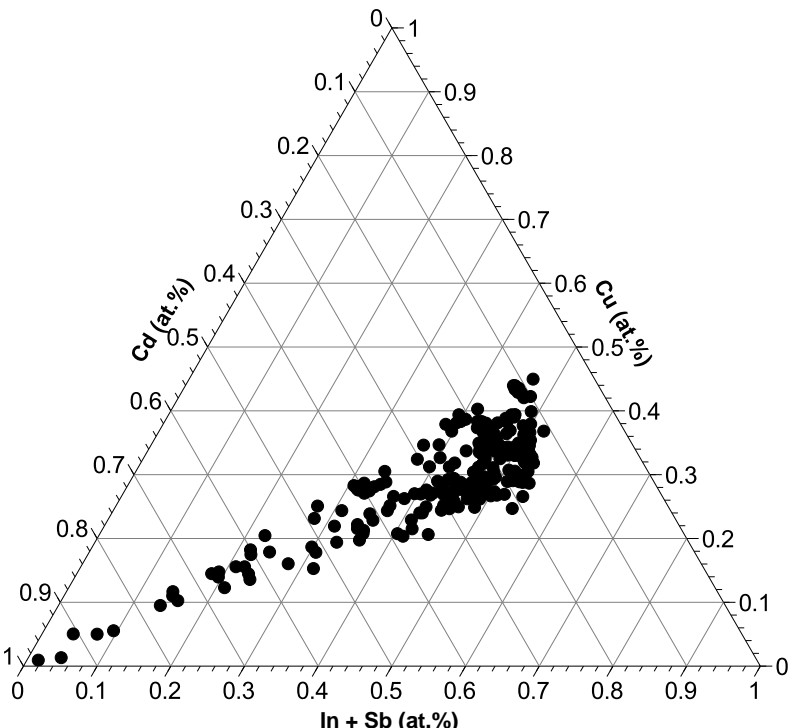

**Figure 4.** Ternary plot of Cu, Cd and In+Sb, all values given in atomic percent, showing a negative trend between Cd on the one side and Cu/(In + Sb) on the other side. For explanations, see text.

This is an indication, that Cd and (Cu + In)/Sb enrichments are mutually exclusive. This is also revealed in the element distribution mappings (see Figure 4).

## 3.2. Selected Sphalerite Element Maps and Scans

To describe the features of the Cu, In and Sb distribution the areas M6 and M14 (see Figures 5 and 6) were selected.

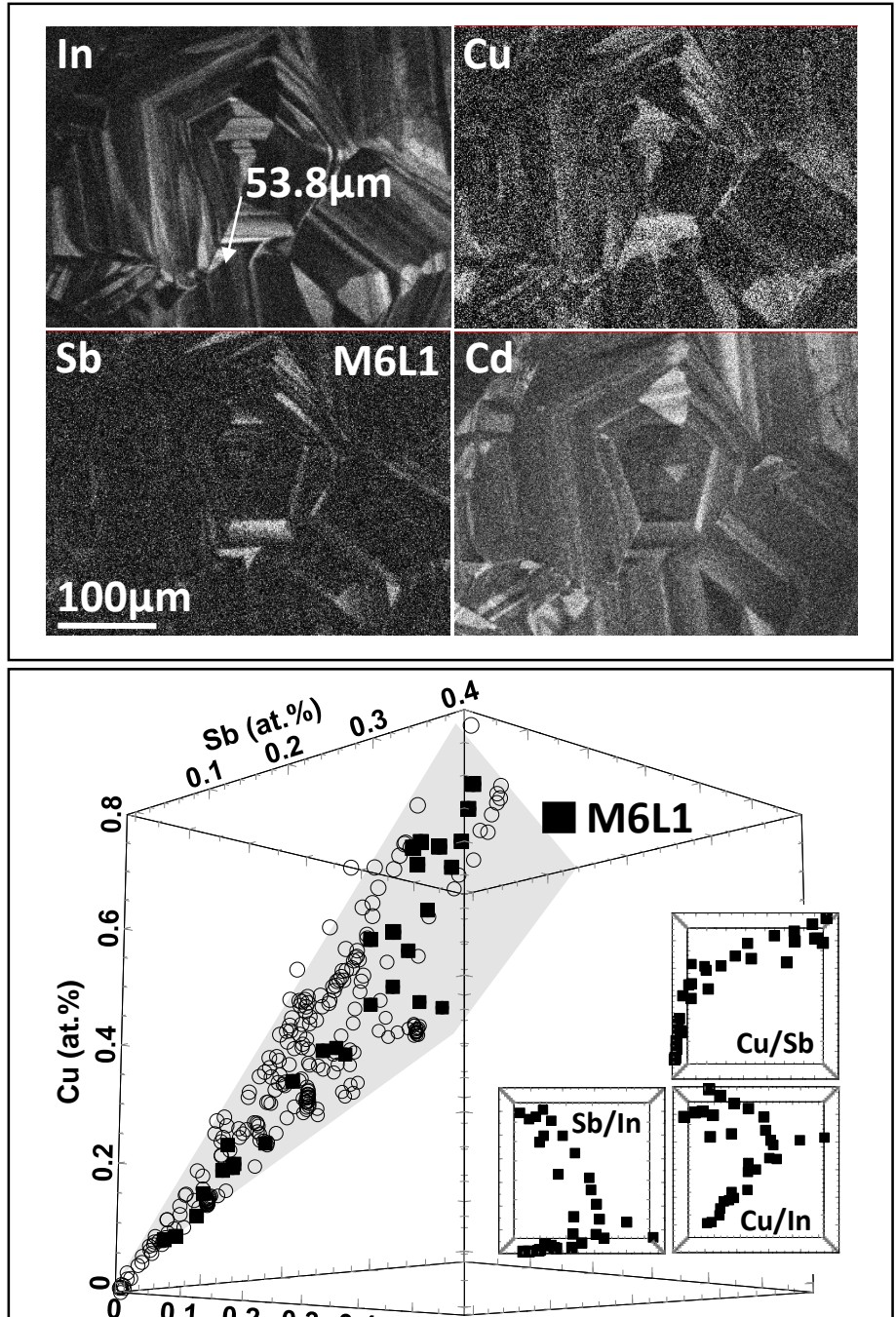

**Figure 5.** Upper four images: element distribution mapping M6 with the direction of element distribution profile M6L1 (upper left, white arrow): Lower diagram: distribution of the measurement points of M6L1 and the correlation of Cu/Sb, Cu/In and Sb/In (small diagrams on right side). For explanations, see text.

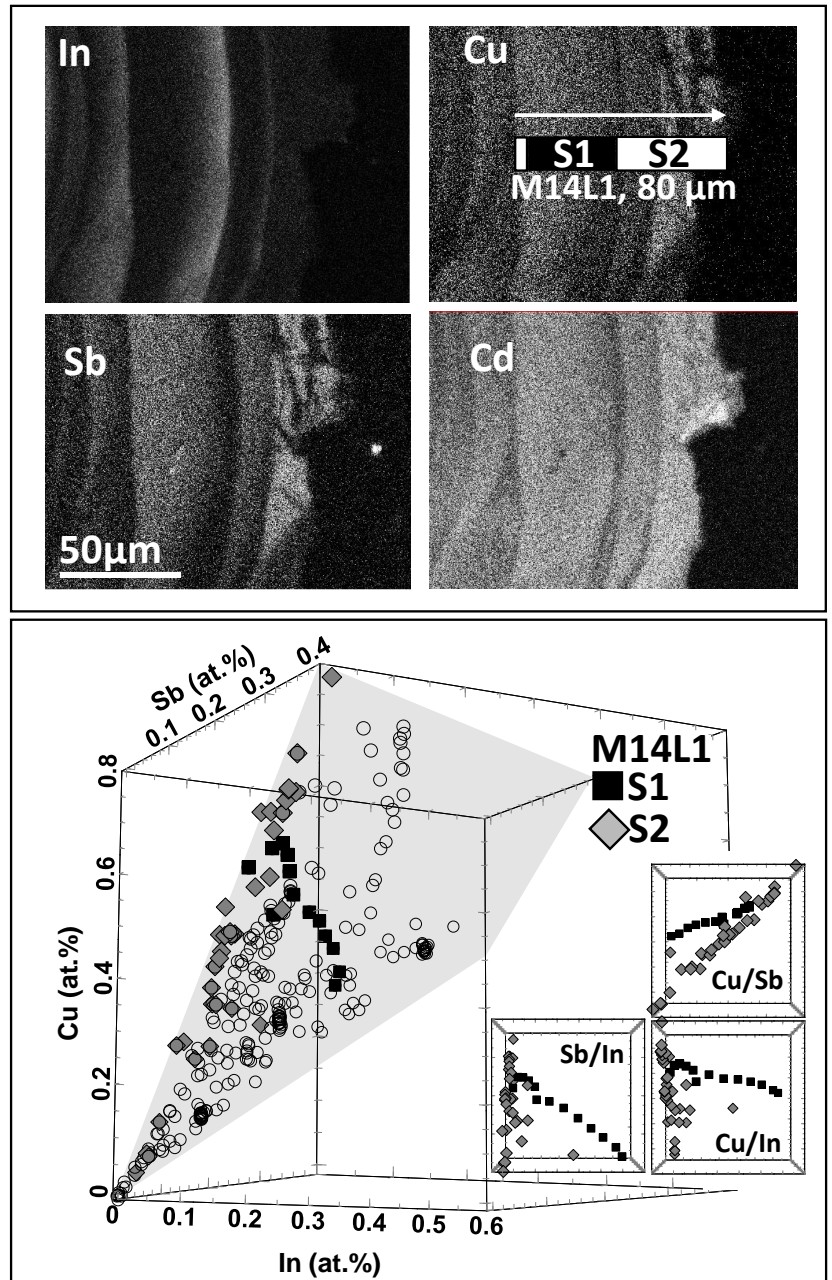

**Figure 6.** Upper four images: element distribution mapping M14 with element distribution profile M14L1 divided into two subscans S1 and S2. Lower diagram: distribution of the measurement points of M6L1 and the correlation of Cu/Sb, Cu/In and Sb/In (small diagrams on right side). For description of the complex correlation situation see text.

In element distribution mapping M6 (Figure 5) the estimated concentration maxima are: 0.74 at.% Cu, 0.20 at.% Cd, 0.52 at.% In, 0.33 at.% Sb. The corresponding data set is listed in the Appendix A (Table A1).

The In and Sb concentration pattern of M6 follows crystallographic planes and forms intermitting layers. The Cd is elevated throughout the analyzed area but also forms an enrichment pattern along crystallographic directions with lines and patches. Nevertheless, the element distribution patterns of Cd and In/Sb are notably inversely correlated. Cu shows a more or less constant concentration within the areas where In and Sb are enriched (Figure 5, upper right, element distribution pattern Cu). In regions with high Cd content the Cu, In and Sb concentrations are decreased. All measurement points

of the element distribution profile M6L1 plot on the plane spanned by the Equation (1). These points are all evenly distributed all over the plane (Figure 5, main diagram below).

Antimony and In substitute each other resulting in a distinct negative correlation. At low concentration of Cu, In correlates with Cu. With rising Cu concentration the In/Cu correlation deteriorates so that only the sum In+Sb continues the line according to Equation (1). In contrast, the correlation between Cu and Sb is pronounced throughout the whole element distribution profile M6L1 (Figure 5, lower x/y diagrams).

In element distribution mapping $M_{14}$, the estimated concentration maxima are: 0.79 at.% Cu, 0.42 at.% Cd, 0.32 wt.% In, 0.39 wt.% Sb. The corresponding data set is listed in the Appendix A (Table A2).

The enrichment pattern of In and Sb (together with Cu) is not as clearly connected with crystallographic planes but rather in the form of irregular elongated patches along the grain boundaries. Cd is again elevated throughout the region and shows none or weak positive correlation with Sb and Cu. The element distribution profile M14L1 (Figure 6, upper right, element distribution pattern Cu) can be divided into two subscans, one that is nearly only a function of Sb (Figure 6, grey diamonds) with low and only slightly varying In concentration and one with distinct negative correlation of In with Sb and Cu (Figure 6, black squares). In the 3D view (Figure 6, main diagram below), the first measurement series plot on the Cu/Sb side, the latter series is crossing the plane from the Cu/Sb side to the Cu/In side of the 3D view. All measurements follow equation 1. Interestingly the Cu/Sb view in Figure 6 (upper small diagram, right) shows two correlations with different slope, represented by grey diamonds and black squares. The latter correlation does not run through the origin of the diagram, indicating, that Sb is exchanged by In in this part of the sphalerite crystal and—for the reason of charge balance—the Cu concentration remains relatively high.

### 3.3. Atom Probe Tomography Results

To gain an understanding of the chemistry on a near-atomic scale, laser assisted APT was used. This allows us to verify the results of the EPMA and extend the analysis to the atomic scale. An overview of the resulting dataset is given in Figure 7a. We captured a volume consisting of ca. 22 million ions, stretching about 300 nm in length. Since the sample experienced a discontinuity during the experiment, possibly a microfracture, we divided the dataset into two sections, leaving out any data that could be influenced.

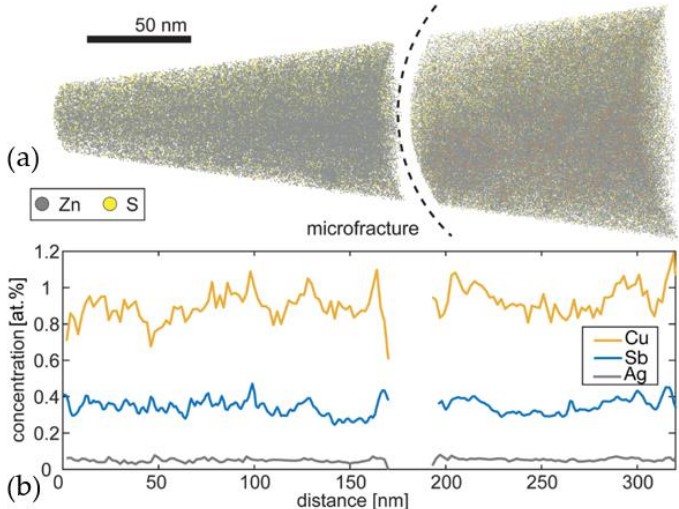

**Figure 7.** (**a**) Overall atom probe reconstruction showing Zn (grey) and S (yellow); (**b**) One-dimensional average concentration profile along the Z-direction showing Cu, Sb and Ag.

In Figure 7a,b, a concentration profile along the measurement axis of the APT sample is shown. This axis is perpendicular to the surface of the EPMA sample. To make this concentration profile and

in fact all following chemical analysis quantitative, the following has to be considered: In APT of minerals, due to the use of laser assisted field evaporation, a high fraction of molecular ions is detected. The detected ions and molecule ions in the presented data are: Zn, Cu, Sb, Fe, Cd, Ag, Sn, $Zn_2S_2$, $Zn_2S$, $FeS_2$, ZnFeS, CdSnS, SSb, ZnS, FeS, $Zn_3S$. This leads to a mass spectrum with a high number of individual peaks, where some share the same mass-to-charge. Nevertheless, the peaks for Cu, Ag and Sb could be uniquely identified based on their natural isotopic abundance. This was not possible for In, whose isotopes ($^{113}In^+$ and $^{115}In^+$) have the same weight as isotopic combinations of $Zn_3S^{2+}$ and for Cu, where only one isotope could unequivocally be identified ($^{63}Cu^+$). However, the peak heights at the positions of $Zn_3S^{2+}$ fit very closely with the natural abundances of the isotopic combinations of $Zn_3S^{2+}$, such that only a very minor amount of In could possibly be present in the investigated volume.

For the quantification of the Cu content in the sample, a correction for the overlap of $65Cu^+$ and $64Zn^+$ had to be made. This was possible by only using $^{63}Cu^+$ for the quantification, which shows no peak overlaps with other elements. We then used the natural abundances of the two isotopes to account for the missing $^{65}Cu^+$. This yields a ratio of Cu:Sb = 2.28:1 in the entire captured volume, which is in good agreement with the EPMA analysis.

Beyond confirming EPMA results, APT allows us to assess the homogeneity of the sphalerite on the nanometer scale. This is presented in Figure 8. In Figure 8a, the atomic distributions of Cu and Sb are shown. In this Figure, only atoms that are tied up in clusters, as determined via Voronoi cell clustering [41] are displayed for clarity. Overall, the clustered atoms represent 5.4% of all detected Cu and Sb atoms, that is, the vast majority of all Cu and Sb atoms are randomly distributed in the APT data. It has to be noted that the detection efficiency of the used instrument is ca. 37%, meaning that very small clusters of only a few atoms would not effectively be reproduced in the 3D reconstruction. In the 3D data of the clustered atoms, it can be seen that mostly smaller clusters, some nanometers in size are present, with one larger cluster about 20 nm in size towards the bottom of the dataset. To gain some insight into the composition of this individual precipitate, we isolated the volume using an iso-concentration surface [42] at 2.5% Cu. (marked with a red arrow in Figure 8b). Relative to this surface, we calculated a concentration vs. distance histogram ('proxigram') [43]. It shows that inside the precipitate, there is an enrichment of Cu, Sb, Ag and to a lesser extent Cd. Fe only shows a relatively weak enrichment.

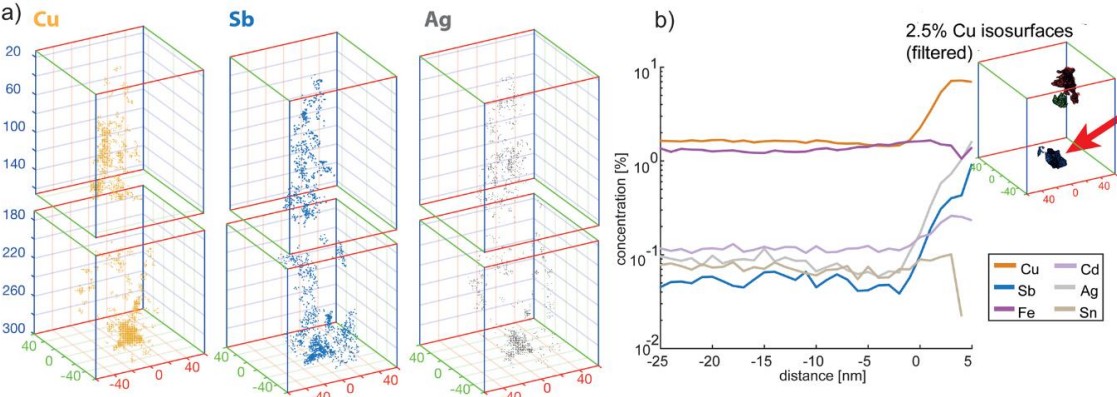

**Figure 8.** Further analysis of the atom probe data shows that the specimen has clusters that are rich in Cu, Sb and Ag relative to the bulk concentration. (**a**) Identified clusters of the elements Cu, Sb and Ag; (**b**) Element distribution profile from outside to inside of the cluster, marked with the red arrow.

## 4. Discussion

### 4.1. General Implications

This study reveals a distinct correlation between Cu, In and Sb in a natural sphalerite. Such a complex incorporation mechanism has not yet been reported in literature (see Equation (1)).

The distinct correlation of the element concentration with crystallographic directions or the formation of crystallization frontiers imply a crystal growth driven by varying physical or chemical properties of the fluid as already described in literature (see Section 3).

Additionally, the results reveal that the influence of the existing boundaries to chalcopyrite (like diffusion of Cu) or to galena (like diffusion of Sb or Ag) is insignificant for the incorporation of In and Sb into sphalerite. This would imply, that the incorporation of In and Sb into sphalerite was finalized prior to the intergrowth of the grains and that during the following stages of the ore-forming process no diffusion processes occurred at the grain boundaries.

Generally, a negative correlation of Cd to In/Sb is observed (Figure 4). Reasons for this pattern could be changes in the fluid chemistry with varying concentrations of Cd. Also, the often-complementary distribution of In and Sb (Figures 5 and 6) imply variations of the fluid from which the sphalerite was crystallizing.

Another explanation could be, that the incorporation of the elements is controlled by different thermodynamic affinities of the elements. If Cd has greater affinity in the sphalerite-structure than In and Sb, an increase of the Cd concentration would lower the incorporation capability of these elements. Considering the element pair In/Sb, Sb could have a greater affinity into the sphalerite-structure than In. This is supported by the fact that Sb nearly always has a positive correlation with Cu, whereas the correlation of Cu and In is not always present (Figures 3, 5 and 6).

### 4.2. Crystal Structural Implications

From the calculated multiple regression function (Equation (1)) over all analyzed element distribution profiles it can be deduced, that the ratio of Cu:In is 1:1 like in $CuInS_2$ (roquesite). This suggests a coupled substitution mechanism, as discussed in literature (see Section 1.3). In contrast, the ratio of Cu:Sb = 1.81:1. This ratio is very near to that of cuprostibite or an As-free form of Watanabeite $(Cu_4(As,Sb)_2S_5)$, $Cu_4Sb_2S_5$. Nevertheless, a compound of this kind could not be identified with EPMA or APT. A coupled substitution like $Cu^{2+} + Cu^+$ in combination with $Sb^{3+}$ as described by [44] would be possible in respect to the stoichiometric point of view but would imply very special redox conditions. An existence of $Cu^{2+}$ within the sphalerite lattice is also contrary to the observations of [16].

The incorporation mechanism of In seems to be well understood and is documented in literature [14]. This is not the case for Sb. Due to the different Cu:Sb ratio, the coupled substitution mechanism discussed for In is not feasible. A first approach to discuss a different behavior of In and Sb is the comparison of the chemical properties. The speciation of In under ambient conditions is always In(III), whereas Sb occurs as Sb(III) and Sb(V). Additionally, Sb can be found in minerals like $CoSb_3$ or $Cu_2Sb$. In these compounds Sb acts as anion ($CoSb_3$) or forms intermetallic structures ($Cu_2Sb$, e.g., [45]). Furthermore, the crystal radii of In and Sb differ considerably (coordination number 4, see Table 2).

**Table 2.** Crystal radii for selected elements on basis interatomic distances in halides and chalcogenides [46], PY—pyramidal. % Diff(Zn)—percentage difference to the radius of $Zn^{2+}$.

| Ion | Coordination | Charge | Crystal Radius (Å) | % Diff (Zn) |
|-----|--------------|--------|--------------------|-------------|
| Zn  | IV           | 2      | 0.74               |             |
| Cu  | IV           | 1      | 0.74               | 0           |
| In  | IV           | 3      | 0.76               | −3          |
| Sb  | IVPY         | 3      | 0.9                | −22         |

As a result, Sb will not fit as well as In into the sphalerite-lattice position of Zn. Furthermore, $Sb^-$ and Cu-containing compounds are likely to have different crystal lattice parameters and bonding types. Finally, all results, calculations and theoretical considerations can be summarized as follows:

If coupled substitution takes place, the compound serving as the basis for this assumption must have a ratio of Cu:Sb = 2:1. There are two possible candidates: an As-free variant of Watanabeite, $Cu_4Sb_2S_5$ and cuprostibite ($Cu_2Sb$).

Watanabeite is an orthorhombic sulfide with unknown symmetry and space group (e.g., [47]) but with totally different cell dimensions, compared with sphalerite. Furthermore, a pure Sb containing form is not yet described in the literature. Because of the substantially different cell parameters incorporation in this way directly into the sphalerite lattice is implausible. Minute exsolution domains smaller than the resolution of the used EPMA would be difficult to detect, because of the similarity of a virtual $Cu_4Sb_2S_5$ phase to $CuSbS_2$.

Cuprostibite is described as an intermetallic compound of the PbFCl-type [48,49]. A description of incorporation of these intermetallic compounds into the sphalerite lattice is not to be found in literature. In direction of c, the tetragonal lattice of cuprostibite has a smaller mismatch of −13% (roquesite: −104%) but in a/b direction the mismatch is substantially larger (26%, roquesite: −2%). By comparing the cell dimensions the mismatch of roquesite is much higher (−112%) compared with 39% in the case of cuprostibite. Considering these parameters, the incorporation of elements directly into the sphalerite lattice based on this compound cannot be completely dismissed. Understanding the substitution mechanism (e.g., coupled substitution) in the case of intermetallic compounds is difficult because the valence electron localization is very variable [49]. The reason is the bonding type mixture with predominant metallic and less ionic bonding. The incorporation of Sb and Cu in this way—directly into the sphalerite lattice—therefore seems to be less probable than the formation of minute clusters consisting of only a few atomic layers in a dimension unresolvable with field emission (FE)-EPMA (minimum 100 nm spot size). X-Ray absorption spectroscopy studies by [50] reveal, that at higher concentration elements like Au can form clusters of $Au_2S$ within the sphalerite lattice. The authors attributed this to the fact that at higher temperatures this metal can be in a metastable solid solution, which can lead to cluster-forming when cooling down.

Investigations in the present study, carried out with (APT) reveal, that the elements Cu, Sb and Ag (Ag unfortunately could not be analyzed with EPMA due to the LLD) indeed are concentrated in clusters (see Figure 8). This observation corroborates the theory of exsolution of these three elements into intermetallic compound clusters at the nanometer-scale. Due to the large analysis volume compared to APT, the EPMA would deliver a bulk chemical result of a pattern of clusters in a sphalerite host crystal. Therefore, the theory of metastable solid solution at high temperatures followed by the formation of clusters upon cooling as reported by [50] seems to be plausible also in this case.

If compounds with a ratio of Cu:Sb = 2:1 would be the basis for incorporation or coupled substitution, the stoichiometric mismatch of the calculated ratio Cu:Sb = 1.81:1 would indicate a lack of Cu. The reason for this might be that a small portion of Sb is incorporated together with Ag. With APT the presence of Ag could be proved in clusters rich in Cu and Sb (Figure 8). Higher Ag concentration in sphalerite seems, in most cases, to be the result of minute inclusions of Ag minerals within the sphalerite crystal (e.g., [9]). A typical mineral previously found in the Harz Mountain vain mineralizations is dyscrasite ($Ag_3Sb$, e.g., [51]).

To test if a monovalent element like Ag can play an important role for the overall correlation of Cu, In and Sb a virtual component $Ag^{*+}$ was calculated via the ionic balance:

$$Ag^{*+} = In\ (at.\%) + Sb\ (at.\%) − Cu\ (at.\%) \tag{3}$$

The values of this component are spreading around zero with a slightly negative value at −0.0053. This proves, that the overall correlation between Cu, In and Sb does not depend on a less concentrated element like Ag.

Figure 6 demonstrates the complexity of the incorporation mechanism with varying domains of In and Sb substitution. The two different correlations for Cu/Sb in the measurement data of M14L1S1 and S2 indicate that two composition domains, one with nearly pure Cu/Sb composition another with Cu/In/Sb composition can occur even in micrometer distances.

A processing route of the sphalerite to extract the Indium was not within the scope of the investigations in this study. Nevertheless, the (re)-processing of a sulfide ore mineral containing material was investigated with the example of a tailing from the Rammelsberg mine site (mentioned in the introduction) [52] (German). In this study the first step was the production of a sulfide concentrate via flotation with aerophine and flotanol CO7. With this method about 70% of the sphalerite could be extracted. The resulting sulfide concentrate was leached with HCl and $H_2SO_4$. The most efficient approach on a laboratory scale to extract the Indium was to use atmospheric leaching and 1 mol/L HCl with 100 g/L solid (unannealed) at 80 °C, 75 L/h air flow and 600 rev/min.

## 5. Conclusions

The results presented in this study strongly suggest that the incorporation mechanisms of In and Sb in sphalerite are different. Whereas In follows the coupled substitution strategy as described in many published articles, Sb seems to form clusters of intermetallic compounds like $Cu_2Sb$.

Further investigations and a verification of this element distribution and behavior could help to develop new models of fluid transport and origin during the ore forming process of the Harz region. The very complex element pattern of Cu, In and Sb of the analyzed sphalerite could be due to a varying fluid chemistry with three basically independent sources for In, Cd and Sb. In this context the missing diffusion processes at the grain boundaries could be interesting as well.

If the affinity of the sphalerite lattice for Sb is higher than the solubility of In, the sphalerite lattice could be controlled by the amount of Sb available during crystallization. This result could be interesting for thermodynamic modeling of sulfides containing trace elements.

The calculated regression function could help to improve the understanding of differences in the behavior of In and Sb and similar elements with respect to incorporation into compatible crystal structures (distribution, clustering, intermetallic behavior).

**Author Contributions:** T.S. conceived the paper, performed the EPMA, the mineralogical characterization and interpretation as well as the literature review in general. C.M. and P.F. conducted the APT (method development, measurement), data reduction (APT) and interpretation (APT) as well as the literature review (APT). W.L. wrote the geological section of the introduction section. All authors have read and agreed to the published version of the manuscript.

**Funding:** P.F. and C.M. acknowledge financial support by the Bavarian Ministry of Economic Affairs and Media, Energy and Technology for the joint projects in the framework of the Helmholtz Institute Erlangen-Nürnberg for Renewable Energy (IEK-11) of Forschungszentrum Jülich. P.F. and C.M. would also like to acknowledge funding by the Deutsche Forschungsgemeinschaft (DFG) via the Cluster of Excellence 'Engineering of Advanced Materials' (project EXC 315).

**Acknowledgments:** We would like to express our gratitude to Kirsten Kempf by providing the first assessment of an extensive selection of Upper Harz Mountain gangue mineralization that simplified the selection of interesting samples. Elmar Plischke from the Institute of Disposal Research is thanked for mathematical considerations and advice. We acknowledge support by Open Access Publishing Fund of Clausthal University of Technology.

**Conflicts of Interest:** The authors declare no conflict of interest.

## Appendix A

**Table A1.** Concentrations (at.%) of $M_6L_1$, determined with EPMA. Zn and S screened with short measurement time to be used for matrix correction.

| Distance (μ) | Zn | S | Fe | Cd | Cu | In | Sb |
|---|---|---|---|---|---|---|---|
| 0 | 48.7 | 50.0 | 1.01 | 0.11 | 0.08 | 0.07 | <0.002 |
| 1.86 | 48.4 | 50.2 | 0.75 | 0.09 | 0.26 | 0.22 | <0.002 |
| 3.71 | 48.5 | 50.3 | 0.68 | 0.07 | 0.22 | 0.18 | <0.002 |
| 5.57 | 48.8 | 50.3 | 0.57 | 0.08 | 0.16 | 0.12 | <0.002 |
| 7.42 | 48.8 | 50.0 | 0.72 | 0.08 | 0.20 | 0.14 | 0.011 |
| 9.28 | 48.6 | 50.2 | 0.67 | 0.07 | 0.25 | 0.17 | <0.002 |
| 11.14 | 48.6 | 50.0 | 0.65 | 0.08 | 0.38 | 0.26 | 0.01 |
| 12.99 | 48.4 | 50.3 | 0.53 | 0.11 | 0.41 | 0.23 | 0.07 |
| 14.85 | 48.0 | 49.9 | 0.78 | 0.20 | 0.65 | 0.09 | 0.31 |
| 16.7 | 47.3 | 50.2 | 1.07 | 0.21 | 0.74 | 0.11 | 0.33 |
| 18.56 | 47.4 | 50.3 | 0.87 | 0.20 | 0.73 | 0.15 | 0.31 |
| 20.42 | 47.4 | 50.4 | 0.81 | 0.20 | 0.70 | 0.19 | 0.27 |
| 22.27 | 47.5 | 50.4 | 0.82 | 0.19 | 0.68 | 0.24 | 0.22 |
| 24.13 | 47.4 | 50.1 | 1.20 | 0.17 | 0.64 | 0.29 | 0.16 |
| 25.98 | 47.7 | 50.1 | 1.06 | 0.16 | 0.58 | 0.30 | 0.13 |
| 27.84 | 47.6 | 50.4 | 0.93 | 0.14 | 0.53 | 0.32 | 0.09 |
| 29.69 | 47.9 | 50.1 | 0.97 | 0.13 | 0.51 | 0.33 | 0.06 |
| 31.55 | 48.0 | 50.1 | 0.99 | 0.11 | 0.44 | 0.31 | 0.02 |
| 33.41 | 48.4 | 50.2 | 0.65 | 0.10 | 0.36 | 0.23 | 0.03 |
| 35.26 | 48.3 | 50.1 | 0.74 | 0.10 | 0.43 | 0.34 | 0.01 |
| 37.12 | 48.1 | 50.1 | 0.60 | 0.10 | 0.55 | 0.52 | 0.01 |
| 38.97 | 48.1 | 50.2 | 0.60 | 0.11 | 0.54 | 0.43 | 0.05 |
| 40.83 | 48.0 | 50.1 | 0.83 | 0.18 | 0.54 | 0.11 | 0.25 |
| 42.69 | 47.6 | 50.4 | 0.88 | 0.20 | 0.63 | 0.03 | 0.32 |
| 44.54 | 47.6 | 50.3 | 0.93 | 0.20 | 0.65 | 0.06 | 0.31 |
| 46.4 | 47.5 | 50.3 | 1.00 | 0.19 | 0.64 | 0.12 | 0.27 |
| 48.25 | 47.7 | 50.2 | 1.09 | 0.17 | 0.56 | 0.18 | 0.17 |
| 50.11 | 48.2 | 50.3 | 1.03 | 0.15 | 0.21 | 0.16 | 0.01 |
| 51.97 | 48.5 | 50.0 | 1.16 | 0.15 | 0.12 | 0.12 | <0.002 |
| 53.82 | 48.6 | 50.2 | 0.97 | 0.15 | 0.09 | 0.09 | <0.002 |

**Table A2.** Concentrations of $M_{14}L_1$ (at.%), with subscans $S_1$ (marked in grey) and $S_2$, determined with EPMA. Zn and S screened with short measurement time to be used for matrix correction.

| Distance (μ) | Zn | S | Fe | Cd | Cu | In | Sb |
|---|---|---|---|---|---|---|---|
| 0 | 47.6 | 50.2 | 1.09 | 0.21 | 0.63 | 0.01 | 0.33 |
| 1.64 | 47.4 | 50.3 | 1.22 | 0.21 | 0.63 | 0.02 | 0.33 |
| 3.29 | 47.2 | 50.4 | 1.21 | 0.21 | 0.61 | 0.02 | 0.32 |
| 4.93 | 47.3 | 50.6 | 0.96 | 0.20 | 0.59 | 0.02 | 0.31 |
| 6.57 | 47.4 | 50.3 | 1.21 | 0.20 | 0.57 | 0.02 | 0.29 |
| 8.21 | 47.6 | 50.1 | 1.33 | 0.19 | 0.49 | 0.02 | 0.25 |
| 9.86 | 47.9 | 49.0 | 2.03 | 0.18 | 0.53 | 0.03 | 0.23 |
| 11.5 | 47.5 | 50.2 | 1.33 | 0.19 | 0.56 | 0.04 | 0.26 |
| 13.14 | 47.7 | 50.1 | 1.14 | 0.19 | 0.57 | 0.06 | 0.26 |
| 14.78 | 47.7 | 50.0 | 1.24 | 0.19 | 0.55 | 0.07 | 0.25 |
| 16.43 | 47.8 | 49.9 | 1.20 | 0.19 | 0.53 | 0.09 | 0.23 |
| 18.07 | 47.9 | 49.8 | 1.34 | 0.20 | 0.47 | 0.09 | 0.19 |
| 19.71 | 47.8 | 49.9 | 1.31 | 0.19 | 0.51 | 0.13 | 0.18 |
| 21.36 | 47.9 | 50.0 | 1.10 | 0.18 | 0.49 | 0.16 | 0.17 |
| 23 | 47.9 | 49.9 | 1.16 | 0.18 | 0.49 | 0.20 | 0.13 |

**Table A2.** *Cont.*

| Distance (μ) | Zn | S | Fe | Cd | Cu | In | Sb |
|---|---|---|---|---|---|---|---|
| 24.64 | 47.9 | 49.7 | 1.42 | 0.18 | 0.47 | 0.24 | 0.10 |
| 26.28 | 47.8 | 49.9 | 1.22 | 0.18 | 0.46 | 0.27 | 0.08 |
| 27.93 | 48.0 | 49.9 | 1.12 | 0.17 | 0.43 | 0.30 | 0.05 |
| 29.57 | 48.2 | 49.7 | 1.21 | 0.13 | 0.41 | 0.32 | 0.02 |
| 31.21 | 48.6 | 50.0 | 0.72 | 0.10 | 0.32 | 0.19 | 0.03 |
| 32.85 | 48.8 | 49.8 | 0.89 | 0.12 | 0.23 | 0.06 | 0.07 |
| 34.5 | 48.7 | 50.2 | 0.49 | 0.15 | 0.31 | 0.04 | 0.15 |
| 36.14 | 48.5 | 50.0 | 0.74 | 0.19 | 0.42 | 0.01 | 0.21 |
| 37.78 | 48.5 | 50.0 | 0.71 | 0.18 | 0.42 | 0.01 | 0.21 |
| 39.43 | 48.4 | 49.8 | 1.03 | 0.18 | 0.39 | 0.02 | 0.20 |
| 41.07 | 48.4 | 50.2 | 0.63 | 0.18 | 0.38 | 0.02 | 0.19 |
| 42.71 | 48.4 | 50.1 | 0.74 | 0.17 | 0.37 | 0.02 | 0.18 |
| 44.35 | 48.4 | 50.2 | 0.66 | 0.17 | 0.34 | 0.03 | 0.16 |
| 46 | 48.5 | 49.8 | 1.05 | 0.16 | 0.32 | 0.05 | 0.13 |
| 47.64 | 48.4 | 50.2 | 0.75 | 0.16 | 0.31 | 0.08 | 0.13 |
| 49.28 | 47.9 | 50.4 | 0.73 | 0.18 | 0.46 | 0.07 | 0.24 |
| 50.92 | 47.7 | 50.2 | 0.87 | 0.23 | 0.62 | 0.03 | 0.32 |
| 52.57 | 47.3 | 50.4 | 1.12 | 0.24 | 0.62 | 0.03 | 0.25 |
| 54.21 | 47.3 | 50.4 | 0.85 | 0.25 | 0.79 | 0.02 | 0.39 |
| 55.85 | 48.0 | 50.2 | 0.82 | 0.22 | 0.50 | 0.03 | 0.28 |
| 57.5 | 48.4 | 50.6 | 0.71 | 0.17 | 0.06 | 0.03 | 0.02 |
| 59.14 | 48.3 | 50.7 | 0.80 | 0.18 | 0.03 | 0.03 | <0.002 |
| 60.78 | 47.9 | 50.4 | 1.04 | 0.22 | 0.26 | 0.02 | 0.11 |
| 62.42 | 47.8 | 50.8 | 0.96 | 0.21 | 0.12 | 0.03 | 0.05 |
| 64.07 | 47.2 | 51.0 | 0.92 | 0.22 | 0.42 | 0.03 | 0.20 |
| 65.71 | 47.4 | 50.8 | 1.18 | 0.24 | 0.24 | 0.05 | 0.13 |
| 67.35 | 47.0 | 50.7 | 1.39 | 0.28 | 0.43 | 0.03 | 0.20 |
| 68.99 | 46.6 | 50.7 | 1.66 | 0.37 | 0.47 | 0.02 | 0.20 |
| 70.64 | 46.4 | 51.0 | 1.30 | 0.33 | 0.60 | 0.02 | 0.29 |
| 72.28 | 46.3 | 51.4 | 0.96 | 0.28 | 0.68 | 0.02 | 0.34 |
| 73.92 | 46.6 | 51.1 | 1.28 | 0.39 | 0.42 | 0.01 | 0.19 |
| 75.57 | 47.0 | 50.9 | 1.39 | 0.42 | 0.25 | 0.01 | 0.11 |
| 77.21 | 47.2 | 51.6 | 0.90 | 0.23 | 0.07 | 0.03 | 0.02 |
| 78.85 | 47.8 | 51.1 | 0.86 | 0.18 | <0.01 | 0.01 | <0.002 |
| 80.49 | 47.9 | 51.5 | 0.46 | 0.15 | <0.01 | 0.01 | <0.002 |

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
