# Peer review of "Indium and Antimony Distribution in a Sphalerite from the “Burgstaetter Gangzug” of the Upper Harz Mountains Pb-Zn Mineralization"

_minerals, doi:10.3390/min10090791_

Round 1
Reviewer 1 Report
Comments on manuscrist minerals-918676-peer-review-v1 entitled ‘Indium and antimony distribution in a sphalerite from the “Burgstaetter Gangzug” of the Upper Harz Mountains Pb-Zn mineralization’ coauthored by Schirmer et al., submitted to Minerals
Schirmer et al. present a new observation on In, Sb, and Cu correlative relationship in a natural sphalerite sample using EMPA and APT techniques, suggesting that a fundamental substitution of these elements to Zn/Cu in the crystal structure and the concentration zonation was recorded by sphalerite growth from hydrothermal fluids, likely reflecting the physiochemical condition of the mineral formation (together with the crystal chemistry) but not affected by elemental diffusion. Thus, this study is suitable to the scope of Minerals, which if published would attract a broad range of readership from mineral industry, academia, and others, who concern about the critical materials like In. Although the manuscript (ms) was well written and organized, numerous minors are picked up and marked on the annotated PDF ms for authors’ consideration when a revision is made to improve the presentation of this paper. A few issues or suggestions need considered by authors below.
- Check the atomic ratios in equation 1. If it is corrected on the basis of multilinear regression of the EMPA data, Cu:Sb atomic ratio should 1:1.81 rather than 2:1.09 as suggested in Line 230. I note that Lines 355-356 mentioned the ratio is likely attributed to watanabeite, but SEM and ATP imaging did not show the presence of this mineral inclusions at nanometer scale!
- The reader expects a plausible solution from this study of how to effectively extract In from sphalerite via what processes economically. Any proposal on this subject would be welcome, especially from mineral processing community.
- Sequential order of appearance of equations should start from 1, then 2, etc. However, the ms started from equation 2. Why is so?
- Appendix is suggested to serve as an electronic supplementary material. Or, place it after the reference section if it kept.
Finally, I would like to see this paper published in the journal because it is a great contribution to mineral study.

Author Response
First of all we would like to thank you for elaborately reviewing our paper.
This was very helpful and we learned a lot in terms to improve our future work. We hope the final revision is according to your suggestions and our manuscript can finally be accepted.
All suggestions marked in the pdf provided are addressed (with markups in revised document), additionally I have some comments to the revision:
Check the atomic ratios in equation 1. If it is corrected on the basis of multilinear regression of the EMPA data, Cu:Sb atomic ratio should 1:1.81 rather than 2:1.09 as suggested in Line 230.
Changed: The equation Cu = 0.98In + 1.81Sb + 0.03 translates to a ratio of Cu : Sb = 1.81 : 1
I note that Lines 355-356 mentioned the ratio is likely attributed to watanabeite, but SEM and ATP imaging did not show the presence of this mineral inclusions at nanometer scale!
Changed: This ratio is very near to that of cuprostibite or an As-free form of Watanabeite (Cu4(As,Sb)2S5), Cu4Sb2S5. Nevertheless, a compound of this kind could not be identified with EPMA or APT.
The reader expects a plausible solution from this study of how to effectively extract In from sphalerite via what processes economically. Any proposal on this subject would be welcome, especially from mineral processing community.
Changed: Added Text:
A processing route of the sphalerite to extract the Indium was not within the scope of the investigations in this study. Nevertheless, the (re)-processing of a sulfide ore mineral containing material was investigated with the example of a tailing from the Rammelsberg mine site (mentioned in the introduction) [52, German]. In this study the first step was the production of a sulfide concentrate via flotation with aerophine and flotanol CO7. With this method about 70% of the sphalerite could be extracted. The resulting sulfide concentrate was leached with HCl and H2SO4. The most efficient approach on a laboratory scale to extract the Indium was to use atmospheric leaching and 1 mole/l HCl with 100 g/l solid (unannealed) at 80 °C, 75 l/h air flow and 600 rev/min.
Sequential order of appearance of equations should start from 1, then 2, etc. However, the ms started from equation 2. Why is so?
Changed: Appearance of equation renumbered
Appendix is suggested to serve as an electronic supplementary material. Or, place it after the reference section if it kept.
Changed: Appendix moved to the end
160: Table 1 à Is this essential?
I think yes, it gives information about the method (meas. times, detection limits). Analysts like me are appreciating an overview like this.
170: APT
No change: only mark without comment – I do not understand what to change…
Reviewer 2 Report
This paper presents the results of investigation the geochemistry of sphalerite from the hydrothermal veins of the Upper Harz Mountain. The authors rightly point out that the results of their research can help in solving a number of questions of the In and Sb in hydrothermal deposits. The main attention in the article was directed to the study of the mechanism of Cu, In and Sb incorporation into the sphalerite structure which has not been reported for natural sphalerite. The main conclusions of the authors are based on their own analytical data obtained using modern equipment (electron probe micro-analysis and atom probe tomography) and processed by methods of mathematical statistics (multilinear regression).The data interpretations seem to have been done well, also the data obtained may be valuable for understanding the geochemistry of In and Sb in hydrothermal ore deposits.
So, the paper adds valuable new information concerning our knowledge on the incorporation mechanisms of in the sphalerite under consideration. I thus can recommend the paper for publication in Minerals but with minor revisions.
Comments and noticed typos
Line 75: should be [5-13]
Line 83 equation 2: numbers of equations should be checked
Line 86 Sn is not trivalent cation. Should be explained.
Line 87 should be [e.g. 7, 10, 13, 14, 17-20].
Line 92 According MMA according to the MMA, the formula of sakuraiite is
(Cu,Zn,Fe)3(In,Sn)S4 (see mindat.org)
Line 101 should be [14, 27-30]
Line 126 should be 0.4-0.7 % Cd
Line 229 Equation 1: numbers of equations should be checked
In general, authors should carefully edit the manuscript to correct typos, references, and so on.
Author Response
First of all we would like to thank you for elaborately reviewing our paper.
This was very helpful and we learned a lot in terms to improve our future work. We hope the final revision is according to your suggestions and our manuscript can finally be accepted.
All suggestions provided are addressed (with markups in revised document), additionally I have some comments to the revision:
Line 86 Sn is not trivalent cation. Should be explained.
Changed: Added Text:
Due to the described coupled substitution mechanism (equation 1) the concentrations of the monovalent (Cu, Ag) and trivalent cations (Ga, In, Sb) and even tetravalent cations (Ga,Sn) can be correlated.
4Zn2+ ↔ X+ + Y3+ + Z4+ Equation 1
In this general equation, X represents monovalent elements like Cu or Ag, Y represents In, Sb and Z represents Ga or Sn. Data with correlations of this type is published by various authors [e.g. 7, 10, 13, 14, 17 – 20].